# Utilization of Food and Agricultural Residues for a Flexible Biogas Production: Process Stability and Effects on Needed Biogas Storage Capacities

**Ervin Saracevic** [1,†], **Susanne Frühauf** [2,†], **Angela Miltner** [1], **Kwankao Karnpakdee** [2], **Bernhard Munk** [3], **Michael Lebuhn** [3], **Bernhard Wlcek** [2], **Jonas Leber** [2], **Javier Lizasoain** [2], **Anton Friedl** [1], **Andreas Gronauer** [2] and **Alexander Bauer** [2,*]

1    Institute of Chemical, Environmental and Bioscience Engineering, TU Wien, 1060 Vienna, Austria
2    Department of Sustainable Agricultural Systems, Institute of Agricultural Engineering,
     University of Natural Resources and Life Sciences, 1190 Vienna, Austria
3    Bavarian State Research Center for Agriculture, Central Department for Quality Assurance and Analytics,
     85354 Freising, Germany
*    Correspondence: alexander.bauer@boku.ac.at; Tel.: +43-1-47654-93135
†    This research was conducted in equal part by Ervin Saracevic and Susanne Frühauf.

**Abstract:** Biogas plants can contribute to future energy systems' stability through flexible power generation. To provide power flexibly, a demand-oriented biogas supply is necessary, which may be ensured by applying flexible feeding strategies. In this study, the impacts of applying three different feeding strategies (1x, 3x and 9x feeding per day) on the biogas and methane production and process stability parameters were determined for a biogas plant with a focus on waste treatment. Two feedstocks that differed in (1) high fat and (2) higher carbohydrate content were investigated during semi-continuous fermentation tests. Measurements of the short chain fatty acids concentration, pH value, TVA/TIC ratio and total ammonium and ammonia content along with a molecular biology analysis were conducted to assess the effects on process stability. The results show that flexible biogas production can be obtained without negative impacts on the process performance and that production peaks in biogas and methane can be significantly shifted to another time by changing feeding intervals. Implementing the fermentation tests' results into a biogas plant simulation model and an assessment of power generation scenarios focusing on peak-time power generation revealed a considerable reduction potential for the needed biogas storage capacity of up to 73.7%.

**Keywords:** semi-continuous fermentation; anaerobic digestion; food waste; biogas; process simulation; biogas storage

## 1. Introduction

Biomass-based power plants show great potential to serve future energy systems' needs. The storability of the energy carrier, whether in gaseous, liquid or solid form, qualifies these plants for demand-oriented energy supply. The implementation of globally determined climate protection goals led to a significant increase in power plants with fluctuating power generation in the energy system, primarily in wind and photovoltaic power plants. Additional flexibility in the energy system is needed to ensure sufficient balancing of power supply and demand [1].

Most biogas plants operated in Europe are currently designed as baseload power plants with the continuous conversion of biogas to electricity and heat in combined heat and power (CHP) units and an operating time of usually over 8000 full-load hours per year. The combined electrical and thermal efficiency of energy conversion from biogas to power and heat with CHP units is about

85–90% [2]. Several European countries have implemented financial incentives for promoting flexible power generation with biogas plants, mostly in the form of feed-in premiums. A summary of feed-in tariffs, premiums and tenders to promote electricity from biogas in the EU-28 can be found in Pablo-Romero et al. [3]. A demand-oriented biogas supply is necessary for flexible power generation with biogas plants, which can be achieved by biogas storage or flexible biogas production concepts [4]. Biogas can either be stored on-site at the biogas plant using internal or external biogas storages devices or in the gas grid after a biogas upgrading step, where carbon dioxide and impurities are removed. Biogas storage concepts entail additional financial expenditure, due to investment and maintenance costs.

A demand-oriented biogas supply can also be achieved by flexible biogas production. Concepts for flexible biogas production include changes to feeding material and intervals, as well as changes to organic workload and storage of intermediate substances [5]. Biogas production can be manipulated by varying feeding times and regulating the amount of feedstock. Increased biogas production can be achieved shortly after the feeding event, while biogas production is reduced during non-feeding periods. Lv et al. [6] studied the effects of two different feeding strategies (substrate was fed once and twice per day with the same organic loading rate (OLR)) on anaerobic digestion using maize silage as the feedstock. The results suggested that the biogas amount produced after feeding, when feedstock was fed once per day, was higher than when it was fed twice per day. However, the two feeding regimes produced similar amounts of biogas each day. Furthermore, a higher dynamic of volatile fatty acids (VFAs), pH and methanogenesis during the 1x feeding strategy was observed, while the process performance of the 2x feeding strategy was quite constant.

Another study of flexible biogas production was performed by Mulat et al. [7]. Three different feeding strategies were applied: Continuously stirred tank reactors (CSTRs) were fed with distiller's dried grain with solubles (DDGS) under mesophilic condition (38 °C). The reactors showed lower biogas and methane production rates during a more frequent feeding strategy. Another finding was that the methanogenic, as well as the bacterial community, did not change within the different feeding regimes. However, long-term process stability was not negatively affected. Alterations to the feeding management were also applied to CSTRs fed with maize silage, sugar beet silage and cattle slurry in the study by Mauky et al. [8]. Different feedstock mixtures were fed while varying the feeding regime (from 1x to 6x per day). By changing the feeding management at high OLR up to 6 kg volatile solids (VS) m$^{-3}$ d$^{-1}$, the daily content of methane and carbon dioxide, acid concentrations and pH value were shown to alternate after feeding, which is similar to the findings of Mulat et al. However, the overall long-term process stability was also not negatively affected [7].

Appropriate process diagnosis requires quantitative benchmarks for efficient and disturbed process states, and early warning data of process failure allows for taking countermeasures in a timely manner. Conventional determinants, such as the methane yield, total volatile acids/total inorganic carbonate (TVA/TIC), fatty acid profile, total solids (TS), volatile solids (VS), and $NH_3$-N proved to be most meaningful for this purpose, particularly in combination with more recently-developed ecophysiological molecular biology parameters, such as the metabolic quotient (MQ) of methanogenic archaea and transcript/gene ratios (T/G or cDNA/DNA) of key enzymes. These enable assessing the physiological state of distinct microbial guilds and can signal early warnings of process acidification even weeks before warnings from conventional determinants occur [9,10]. The methanogens and syntrophic bacteria are the most sensitive guilds in biogas processes, and methanation failure is typically the primary reason for process disturbance. Analyzing the metabolic performance of methyl coenzyme M reductase (Mcr, Mrt), the key enzyme of methanogenesis and only found relevant in methanogenic archaea [11], is, therefore, most meaningful and suitable for microbial process diagnosis and the early warning of acidification. Several quantitative real-time PCR-based (qPCR) assays targeting the two alleles of subunit A (*mcr*A and *mrt*A or *mcr*A$_{1,2}$) were developed and evaluated [12], and were applied to determine the T/G$_{mcrA1,2}$ ratio and the MQ in the current experiments.

To predict the behavior of a dynamically operated biogas plant including biogas storage requirements, modelling and process simulations are appropriate tools that were applied in previous studies. Grim et al. [13] developed a dynamic biogas plant model and combined it with the well-accepted Anaerobic Digestion Model No. 1 (ADM1) [14] to assess the technical and economic effects of flexible plant operation. They demonstrated that biogas storage requirements could be reduced by well-timed feeding management. O'Shea et al. [15] used kinetic modelling to determine optimal feeding times at a biogas plant with combined biomethane production and electricity generation by a CHP unit. A model predictive control was developed by Mauky et al. [16] to determine optimal feeding schedules in the context of demand-oriented electricity generation using a simplified ADM1 model. Barchmann et al. [17] used the same simplified model for economic comparison of continuous and flexible biogas production in terms of required biogas storage capacity and the overall economic viability of demand-oriented power generation.

The effects of the feeding management on biogas plants with a focus on biomethane production and waste treatment have not yet been thoroughly studied. Thus, the aim of this study is to investigate the influence of flexible feeding strategies on the biogas production and process stability of an Austrian biogas plant for waste treatment with combined biomethane and flexible power generation. In this study, laboratory scale semi-continuous experiments were conducted while varying the feeding intervals (1x, 3x and 9x per day). Two substrate mixtures of the investigated biogas plant were used by applying two different OLRs. The process parameters were analyzed and a microbiological analysis, based on $mcrA_{1,2}$ genes and MQ, was carried out to determine the effects on the process stability. The experimental results were implemented in a process simulation model, and three different scenarios of power generation during peak times were assessed to investigate the effects on the biogas storage capacities of the biogas plant.

## 2. Materials and Methods

### 2.1. Description of the Biogas Plant

In this study, the investigated feedstock was provided from an Austrian biogas plant located in Bruck an der Leitha (Lower Austria). Each year, the biogas plant digests about 34,000 t of organic materials—mainly wastes from the food and fodder industry, kitchen wastes and wastes derived from the agro-industry. The biogas plant consists of three continuously (hourly) fed and stirred main fermenters, which each have a volume of 3000 $m^3$, as well as two post digesters with a volume of 5000 $m^3$ each. The total biogas storage capacity is 4800 $m^3$, which is sufficient to store about 6–7 h of produced biogas under nominal conditions. The sanitization of critical substrates (biogenic waste of animal origin) is done in two tanks, which each have a volume of 25 $m^3$. The fermentation residue (approximately 39,000 $m^3$ per year) is used as a fertilizer for agricultural land. The average OLR of all fermenters is 1.4 kg vs. $m^{-3}$ $d^{-1}$, which is based on the analysis of 20 samples of feedstocks from July to December 2015. The average retention time in 2015 was 73 days.

The biogas plant was initially designed for electricity and heat production by CHP units; however, since 2014, almost the entire amount of produced biogas is converted into biomethane by a biogas upgrading process. The biomethane is fed into the local and national gas grid system. The capacity of the biogas upgrading system is 1000 $m^3$ biogas per hour. In the case of biogas overproduction or upgrading system maintenance, power is produced by two CHPs, which have an electrical capacity of 836 and 526 kW. In 2015, the total biogas production was 4.7 Mio. $m_N^3$ (Methane content of 66%), or more specifically, 1.3 MWh electricity, 1.4 MWh heat and 2.6 Mio. $m_N^3$ biomethane.

### 2.2. Feedstock

#### 2.2.1. Feedstock Analysis

In this experiment, two different types of feedstocks—one with high fat content (feedstock 1) and the other with high carbohydrate content (feedstock 2)—were used. Both feedstocks were obtained from the biogas plant Bruck an der Leitha, and their selection was random. The chemical composition of

feedstock 1 and feedstock 2 lie within the fluctuation range of the chemical composition of 20 analyzed feedstock samples (data not presented) from the biogas plant Bruck an der Leitha. After obtaining the feedstock from the biogas plant Bruck an der Leitha, it was immediately homogenized using the Retsch GRINDOMIX GM 200 knife mill (Retsch GmbH, Haan, Germany), and was then kept at −20 °C to prevent further degradation. The feedstock was first analyzed for dry matter (DM), VS, water content, raw ash (XA), crude protein (XP), cellulose (CEL), hemicellulose (H-CEL) and acid detergent lignin (ADL). Dry matter was determined by drying the feedstock under 105 °C until it obtained a constant weight, while the raw ash was determined at 550 °C [18]. The vs. were then measured by subtracting the amount of raw ash from the total solids. The water content analysis was performed through a titrational method suggested by Karl Fischer, using the HYDRANAL™—Composite 5 and HYDRANAL™—Methanol dry (Sigma Aldrich, St. Louis, MO, USA) as the reagents (titrant). The volumetric Karl Fischer titrator Mettler Toledo V20 (Mettler Toledo, Ohio, USA) was applied. The amounts of neutral detergent fibers (NDF), acid detergent fibers (ADF) and acid detergent lignin (ADL) were measured through the Van Soest method [19]. Ammonium-nitrogen ($NH_4^+$-N) was determined by the Kjeldahl method, using the K-424 digestion unit and the B324 Kjeldahl distillation unit (Büchi Laboratory AG, Flawil, Switzerland). Total crude protein was calculated by multiplying the amount of nitrogen with the factor 6.25 [20]. $NH_3$-N was calculated according to the formula in Hansen et al. [21]. The amount of crude fat was determined by dissolving it with acetone and subsequent drying at the external laboratory (Futtermittellabor Rosenau, Wieselburg-Land, Austria), following the detail described by the Austrian Association of Grassland and Livestock Production (ÖAG, 2017).

The elemental analysis of the feedstocks was conducted by the Microanalytical Laboratory of the University of Vienna (Austria) by using the EA 1108 CHNS-O elemental analyzer (Carlo Erba, Milan, Italy) [22]. The theoretical biogas and methane potential were calculated according to Boyle [23].

### 2.2.2. Feedstock Characterization

The chemical and elemental composition, as well as the theoretical biogas and methane potential of the two feedstocks, are summarized in Table 1.

**Table 1.** Composition of feedstocks 1 and 2, including dry matter (DM), water content, volatile solids (VS), elemental composition, C/N ratio, theoretical biogas and methane potential.

|  |  | Feedstock 1 | Feedstock 2 |
| --- | --- | --- | --- |
| OLR | [kg vs. $m^{-3}$ $d^{-1}$] | 1.4 | 2.15 |
| DM | [% FM] | 15.9 | 9.9 |
| VS | [% FM] | 15.1 | 8.8 |
| DM (KF) | [% FM] | 20.2 | 10.6 |
| Water | [% FM] | 79.8 | 89.4 |
| XA | [% VS] | 5.1 | 11.3 |
| XL | [% VS] | 73.5 | 33 |
| XP | [% VS] | 10.6 | 24.7 |
| CEL | [% VS] | 6.2 | 13.9 |
| H-CEL | [% VS] | 5.1 | 9.9 |
| ADL | [% VS] | 3.0 | 10.5 |
| C | [% VS] | 73.1 | 64.9 |
| H | [% VS] | 12.1 | 10.3 |
| N | [% VS] | 1.6 | 2.9 |
| O | [% VS] | 12.9 | 21.7 |
| S | [% VS] | 0.2 | 0.3 |
| Theoretical biogas potential | [$l_N$ $kg^{-1}$VS] | 1.390 | 1.231 |
| Theoretical methane potential | [$l_N$ $kg^{-1}$VS] | 963 | 796 |

FM, fresh matter; DM, dry matter; VS, volatile solids; KF, Karl Fischer; XA, raw ash; XL, raw fat; XP, crude protein; CEL, cellulose; H-CEL, hemicellulose; ADL, acid detergent lignin; OLR, organic loading rate.

Feedstock 1 was applied to the semi-continuous test at the OLR of 1.4 kg vs. m$^{-3}$ d$^{-1}$. The values of DM and vs. were almost two times higher than those of feedstock 2 (applied at the OLR of 2.15 kg vs. m$^{-3}$ d$^{-1}$), which showed the DM and vs. at 9.9% FM and 8.8% FM. The raw fat contents of feedstock 1 were approximately 2–3 times higher than in feedstock 2. On the contrary, feedstock 2 showed a higher fiber content compared to feedstock 1. Cellulose, hemicellulose and lignin were approximately two times higher than in feedstock 1.

The dry matters determined by the Karl Fischer (KF) titration were two times higher for feedstock 1. These values showed a similar trend to the dry matter performed by the oven dried weight. The amount of crude protein in feedstock 2 was two times higher than in feedstock 1.

The C/N ratios were at 45 and 23 for feedstock 1 and 2, respectively. The C/N ratio is an indication for the nutrient balance of the anaerobic digestion. The optimum C/N ratio for anaerobic digestion was suggested in the ranges between 20 and 30. Li et al. 2011 [24] and Zeshan et al. 2012 [25] reported that the maximum methane potential was achieved by the biodegradable feedstocks at the C/N ratio of 27 and 32. Although some studies have suggested that anaerobic digestion proceeded efficiently at low C/N ratios [26,27], methane production also largely depends on other nutrients, such as fat, crude protein and fibers [28]. In this study, the high amount of raw fat, carbohydrate and protein could indicate that these feedstocks were potentially the high degradable substrates.

### 2.3. Biological Methane Potential Test

The biological methane potential (BMP) test was performed according to VDI 4630 [29] at 37.5 °C and 56 days. The fermenters were connected to a eudiometer with a capacity of 300 ml using an inoculum—sample ratio of 1:3 (based on the vs. content). The mixed inoculum was filtered and diluted with water to 4% DM.

The amount of biogas was monitored daily under the standard condition of 273.15 K and 101.33 kPa, and was reported as normalized liters (l$_N$). The biogas compositions (CH$_4$, CO$_2$, O$_2$, H$_2$S, and H$_2$) were measured by the portable gas analyzer, Dräger X-AM 7000 (Dräger, Luberg, Germany).

Feedstock 1 showed a biological biogas potential of 1142 l$_N$ kg$^{-1}$ vs. and a biological methane potential of 799 l$_N$ kg$^{-1}$ vs. For feedstock 2, the cumulative curves showed the biological biogas potential at 906 l$_N$ kg$^{-1}$ vs. and the biological methane potential at 559 l$_N$ kg$^{-1}$ vs. Therefore, the biological methane potential presented in the BMP test has reached 83% and 70% of their theoretical potential, as shown in Table 1.

For the materials used in the experiments, a rapid degradation during storage can be expected, resulting in a high content of volatile components lost at 105 °C. Therefore, the use of vs. as a basis leads to an overestimation of biogas and methane yields. Therefore, in the experiments, the vs. (KF) was calculated by subtracting the water, as well as the ash (content) of the FM. Thus, the biological biogas/methane potential of feedstock 1 was 899/629 l$_N$ kg$^{-1}$ vs. (KF). Feedstock 2 showed a biological biogas/methane potential of 846/522 l$_N$ kg$^{-1}$ vs. (KF).

### 2.4. Semi-Continuous Flexible Biogas Production

In order to study the influence of flexible feeding systems on biogas and methane production, three different feeding strategies were applied to the laboratory scale fermenter—1x, 3x and 9x feeding per day. The biogas and methane production were measured in a semi-continuous experiment according to VDI 4630 standard. The total volume of the fermenter was two liters, with the working volume of 1.7 liters. The physicochemical characterization of the mixed inoculum has shown a DM content of 2.9% FM and a content of vs. of 1.8% FM, respectively. The pH of the inoculum was at 7.7, with the TVA/TIC ratio of 0.4. The ammonium-nitrogen (NH$_4$$^+$-N) content of the inoculum was at 1.20 g kg$^{-1}$ FM. All of the fermenters were operated under the identical mesophilic condition in the controlled water bath at 37.5 °C. The input and output materials were homogenized, measured and balanced to control the equivalent working volume.

The semi-continuous biogas system was operated for 186 days. The OLR applied in this experiment were simulated from the operating system of the biogas plant at Bruck an der Leitha. The overall timeline for the adaptation and measurement period is shown in Figure 1.

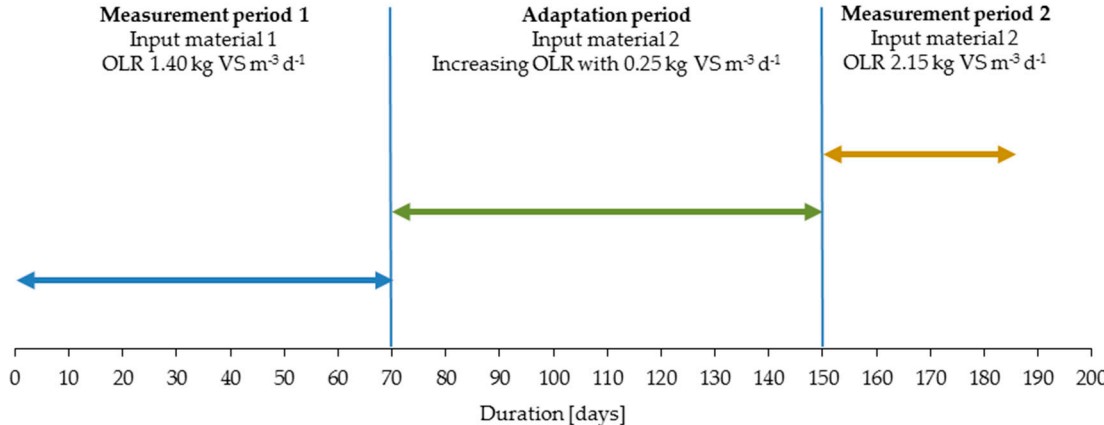

**Figure 1.** The overall timeline for the semi-continuous experiment.

Measurement period 1 was performed at an OLR of 1.40 kg vs. $m^{-3}$ $d^{-1}$. This value represents the average OLR of the Bruck an der Leitha biogas plant during July–December 2015 (mean value of 20 samples in total). In measurement period 1, the fermenters were fed with feedstock 1 for 70 days. Next, feedstock 2 was applied. As described above, feedstock 2 indicated, due to its chemical and elemental composition, a lower theoretical and biological biogas and methane potential compared to feedstock 1. In order to keep the biogas production in both measurement periods at the same level, the OLR was gradually increased with an interval of 0.25 kg vs. $m^{-3}$ $d^{-1}$ until the OLR of 2.15 kg vs. $m^{-3}$ $d^{-1}$ was reached (adaptation period).

During the adaptation period where the OLR raised from 1.4 to 2.15 kg vs. $m^{-3}$ $d^{-1}$, the steady state of biogas production in each operating period was maintained. In this experiment, "steady state" was defined as a constant biogas production that lasted for approximately 2–3 weeks, with the coefficient of variation less than 10% in all feeding strategies after increasing the OLR [30].

The measurement of biogas and methane production is reported at the condition of 273.15 K and 101.33 kPa in normalized liters ($l_N$). The analysis of the biogas composition ($CH_4$, $CO_2$, $O_2$, $H_2S$, and $H_2$) was performed. The parameters for process stability, such as pH, temperature, pressure and the TVA/TIC ratio were also measured throughout the experiment.

*2.5. Process Parameters*

The process parameters, such as short-chain fatty acids (SCFAs), pH, TVA/TIC ratio and ammonium-nitrogen ($NH_4^+$-N) content were analyzed as indicators of process stability during the anaerobic digestion.

Short-chain fatty acids (SCFAs, $C_2$-$C_7$) were investigated using a gas chromatograph (Agilent 6890 N, Agilent Technologies Santa Clara, CA, USA) following the detailed protocol from the previous study [31]. The SCFAs spectrum was measured via suitable extracts in the specific laboratory condition. The detector can analyze acetic acid, propionic acid, hexanoic acid and heptanoic acid. Butyric acid and valeric acid, as well as their isoforms, are also included. The pH value was measured constantly from the beginning of the experiment using the Consort C933 pH meter (Consort, Turnhout, Belgium) and the BlueLine 26 pH electrode (SI Analytics, Mainz; Germany). The TVA/TIC ratio, one of the parameters indicating the appropriate amount of feedstock into the biogas reactor, was conducted via the method described in Buchauer, 1998 [32]. The analysis of ammonium-nitrogen ($NH_4^+$-N) content was also performed according to the Kjeldahl method with the K424 digester and the Büchi B324 distillation instrument B324 (Büchi Laboratory AG, Flawil, Switzerland) [20,21].

## 2.6. Molecular Biology Analyses

The influence of the three feeding strategies on the specific transcriptional activity of methanogenic archaea was studied using quantitative real-time PCR (qPCR) approaches on mRNA and DNA level [10]. Samples were collected from the fermenters and immediately preserved in an RNAlater solution, as suggested by the manufacturer.

Before the extraction of nucleic acids, 500 mg of the samples were homogenized and washed twice with 0.85% KCl to minimize the presence of PCR inhibitors and nucleic acids released from decayed cells [33]. Total DNA and RNA were extracted concomitantly using the Roboklon GeneMatrix DNA+RNA+Protein Extraction Kit in combination with an upstream bead-beating step for most efficient physical and chemical cell disruption and lysis. For this, 200 mg of washed sample was transferred to Lysis Matrix E tubes (MP biomedicals), 300 μL Roti®-Phenol were added, and samples were treated in an MP FastPrep®24 bead-beater (2x 20 s, 5 m/s, with a 5 min pause on ice). After 5 min centrifugation at room temperature (10,000 g), 100 μL of the supernatant (ca. 200 μL) were transferred to a new Eppendorf cup, re-pipetted well, and 300 μL prepared DRP, and 200 μL prepared Lyse All solutions (Roboklon GeneMatrix recipe) were admixed. The solution was transferred to a Roboklon DNA binding spin column, centrifuged (2 min, 16,000 g), and the flow through (RNA + proteins) was transferred to a new 1.5 mL Eppendorf cup. 0.7 vol (420 μL) of molecular grade ethanol was admixed to the flow through: Half of this suspension was transferred to a Roboklon GeneMatrix RNA binding spin column, and DNA and RNA were further extracted following the Roboklon GeneMatrix protocol.

RNA was treated with DNase (TURBO DNA-free™ kit, Ambion) to eliminate contamination by residual DNA. 5 μL of DNA-free RNA was used for reverse transcription (RT) with AffinityScript Multi-Temp RT (Agilent) and primer MeA-i 1435r (see below) according to the manufacturer's protocol for gene-specific primers at 45 °C for 60 min in an Analytik Jena FlexCycler. -RT reactions serving as controls for the presence of residual $mcrA_{1,2}$ DNA were all negative.

The concentration of methanogens and $mcrA_{1,2}$ transcripts (measured as cDNA) in the samples were quantified by qPCR in triplicates using primers MeA-i 1046f and MeA-i 1435r [34] and integrating an internal pre-quantified $mcrA$ standard as described previously [35]. Absolute quantification results considered all dilutions and known losses from sampling to analysis, and plausibility benchmarks [12]. Transcript/gene ratios ($T/G_{mcrA1,2}$) were calculated to display the specific transcriptional activity and the MQ to assess the metabolic state of the methanogens [9] in a given sample.

## 2.7. Process Simulation of Biogas Plant

A simulation flowsheet was developed in the process simulation program IPSEpro version 7.0 to assess the effects of flexible feeding strategies on the needed biogas storage capacities of the biogas plant during flexible power generation and continuous biomethane production. The program's simulation allows for the creation of dynamic simulations, whereby an equation-orientated approach and dynamic time steps are used to calculate flowsheets. A simplified scheme of the developed flowsheet is shown on the left side of Figure 2. It consists of a biogas storage model, models for biogas upgrading and two CHP unit models. A detailed description of the models used in this study can be found in Saracevic et al. [36].

Equations for mass and energy balance were implemented in the flowsheet models. Biogas flow rate and methane concentration can be defined in the inlet flow stream of the biogas storage. The capacity of the investigated biogas plant's biogas storage is 4800 $m^3$, and the maximum biogas storage level in the simulations was set to be 3800 $m^3$ (approximately 79.2% of the total storage capacity). The remaining 1000 $m^3$ of the biogas storage volume was assumed to be a reserve in the case of malfunctions in the biogas upgrading units.

The biogas that is stored in the biogas storage can either be transferred to the biogas upgrading units for biomethane production or to the CHP units for power and heat generation. Continuous biomethane production was assumed in the simulations, which is necessary, due to gas delivery contracts of the biogas plant. The biogas flow rate to the biogas upgrading units was set to be constant

over each simulated day, whenever the biogas storage was not empty or full, and was dependent from the amount of produced biogas. If the biogas storage was full, the biogas flow rate to the biogas upgrading units was increased to the inlet flow rate of the biogas storage.

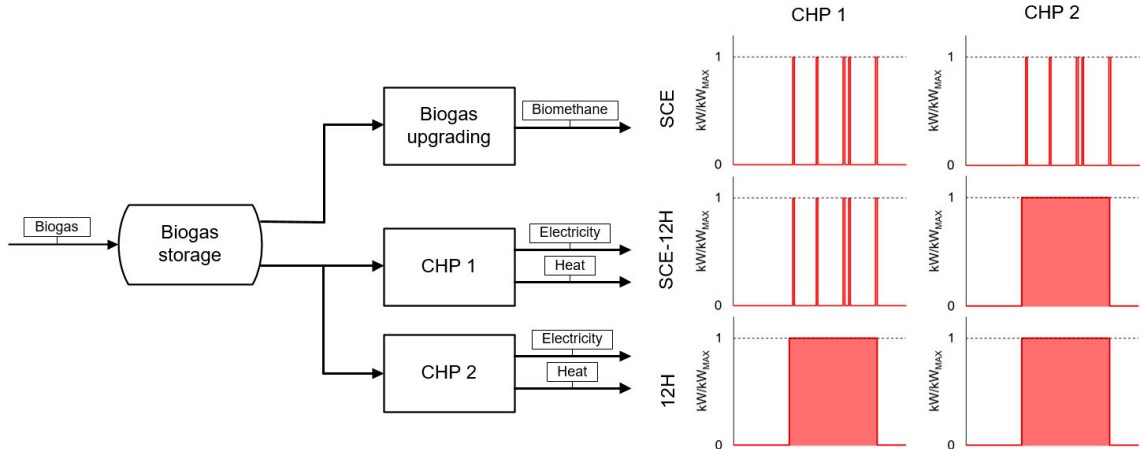

**Figure 2.** Left: Flowsheet of developed process simulation model; Right: Investigated scenarios for power generation.

The biogas upgrading units were modelled as simple black box models (splitters), and the model parameters were set according to regular measurements of the biomethane quality at the biogas plant; the average methane concentration in the investigated period was 98.1% v/v. Functions for electric and thermal efficiency were implemented in the CHP unit models: A detailed description of the models can be found in Wöss et al. [37]. CHP 1 and 2 have electrically-rated outputs of 836 kW and 526 kW, respectively. Timetables for power generation can be defined in the CHP unit models.

Control energy reserves are needed to keep the electricity grid frequency constant at 50 Hz, and they are organized by transmission system operators (TSO). Control energy reserves are activated in the case of an unexpected deviation in the grid frequency, and they are distinguished as negative and positive reserves. If a power plant participates in the market for positive control energy reserves, then the TSO is empowered to increase the power generation of the plant on demand. Control energy reserves are furthermore divided in primary, secondary and tertiary reserves, depending on the required reaction time of the power generation unit and maximum activation time. Secondary control energy reserves must be activated within five minutes after the TSO's signal and can be activated for a maximum of fifteen minutes per activation. A more detailed description of the markets for control energy can be found in literature, i.e., in Panos [38].

The right side of Figure 2 shows the three scenarios for power generation that were assessed in this study. Participation in the market for secondary positive control energy reserves and continuous power generation during peak times (08:00–20:00 on workdays) was assumed in these scenarios. Scenario SCE of this study assumed that both CHP units are used to provide positive secondary control energy reserves during peak times. In scenario SCE-12H, CHP 1 is used to provide positive control energy reserves, and CHP 2 is used for continuous power generation during peak times. In scenario 12H, both CHP units are used to generate power during peak times continuously. It was assumed in all of the simulated scenarios that power is generated according to the power generation schedule, while biogas is continuously upgraded to biomethane.

One month of plant operation was simulated for each scenario. Data published by the Austrian TSO [39] and on the platform for allocation of control energy reserves [40] was used to determine the times when control energy reserves were activated (an energy price of 200 € MWh$^{-1}$ was assumed). Data from January 2017 were used in this work, as this was a month when control energy reserves were activated very frequently. An activation time of three minutes per call-off order was assumed. Biogas flow rates and methane concentrations of the three investigated feeding regimes were defined

in the simulations, and biogas storage levels were compared to continuous feeding of feedstock to assess the reduction potential of the needed biogas storage capacity.

## 3. Results

### 3.1. Semi-Continuous Flexible Biogas Production

Three different feeding strategies (feeding 1x, 3x and 9x per day) were applied to assess their effects on the process performance of a semi-continuous biogas system. Additionally, two different feedstocks were used at an OLR of 1.4 kg vs. $m^{-3}$ $d^{-1}$ for feedstock 1 and 2.15 kg vs. $m^{-3}$ $d^{-1}$ for feedstock 2. The performance of biogas and methane production was analyzed hourly and daily. Figure 3 (a. to c.) illustrates the average hourly biogas production ($l_N$ $m^{-3}$ $h^{-1}$) of the three feeding strategies over a period of 24 h.

For the 1x feeding strategy (Figure 3a), biogas production from both feedstocks was immediately increased within the first hour after the feeding event, which took place 25 minutes before hour 1. The maximum was achieved at hour 2 with a biogas production of 86 and 82 $l_N$ $m^{-3}$ $h^{-1}$ and a methane production of 61 and 63 $l_N$ $m^{-3}$ $h^{-1}$ for feedstocks 1 and 2, respectively. These values make up 128% of the average daily biogas production for both feedstocks. After this peak, the production rate decreased until the next feeding event. The decline was initially more rapid compared to the decline during the last two-thirds of the 24-h period. However, a high amount of biogas and methane production was recorded during the first eight hours. At the end of the 24-h period, the biogas production was at 66% for feedstock 1 and at 68% for feedstock 2 compared to their highest levels at hour 2.

Figure 3b presents the results from the 3x feeding strategy, where fermenters were fed 25 minutes before hour 1, hour 5 and hour 9. Both feedstocks showed a similar pattern of biogas production. Compared to the 1x feeding strategy, the fast increase in biogas and methane production did not occur after the first feeding event. The highest level of biogas production was achieved at about hour 3 after the first feeding event, whereas the highest level was already reached within one hour after the second and third feeding event. However, a similar deterioration rate of biogas production was measured after each feeding event. Both feedstocks had their overall peaks of biogas production at hour 10, where the data show a biogas and methane production of 81 and 57 $l_N$ $m^{-3}$ $h^{-1}$ for feedstock 1 and 82 and 51 $l_N$ $m^{-3}$ $h^{-1}$ for feedstock 2. These values constitute 122% and 129% of their average daily biogas production. From hour 11, biogas production decreased continuously until the next feeding event. At the end of the 24-h period, the biogas production capacity was at 73% for feedstock 1 and at 64% for feedstock 2 compared to their highest capacities at hour 10.

The biogas production for the 9x feeding strategy is shown in Figure 3c, suggesting a continuous increase in biogas production up to hour 10. The first feeding event started 30 min before hour 1, proceeded at hourly feeding intervals and included a feeding break at hour 5. The biogas and methane production patterns from both feedstocks were similar. Biogas and methane production were gradually accumulated until hour 4, which was followed by a slight decrease at hour 5, due to the feeding break. Thereafter, the biogas and methane production further increased until hour 10, where the biogas production peaks for both feedstocks were registered (115% for feedstock 1 and 121% for feedstock 2, compared to the average daily biogas production rate). Feedstock 1 achieved at its maximum point a biogas and methane production of 74 and 52 $l_N$ $m^{-3}$ $h^{-1}$, whereas the maximum point was 77 and 48 $l_N$ $m^{-3}$ $h^{-1}$ for feedstock 2. At the end of the 24-h period, the biogas production capacity was 78% for feedstock 1 and 71% for feedstock 2 compared to their highest production rates at hour 10. The slight increases in biogas and methane production describe the different behavior of a more frequent feeding strategy compared to the more rapid increases in biogas production that occurred in the 1x and 3x feeding strategies.

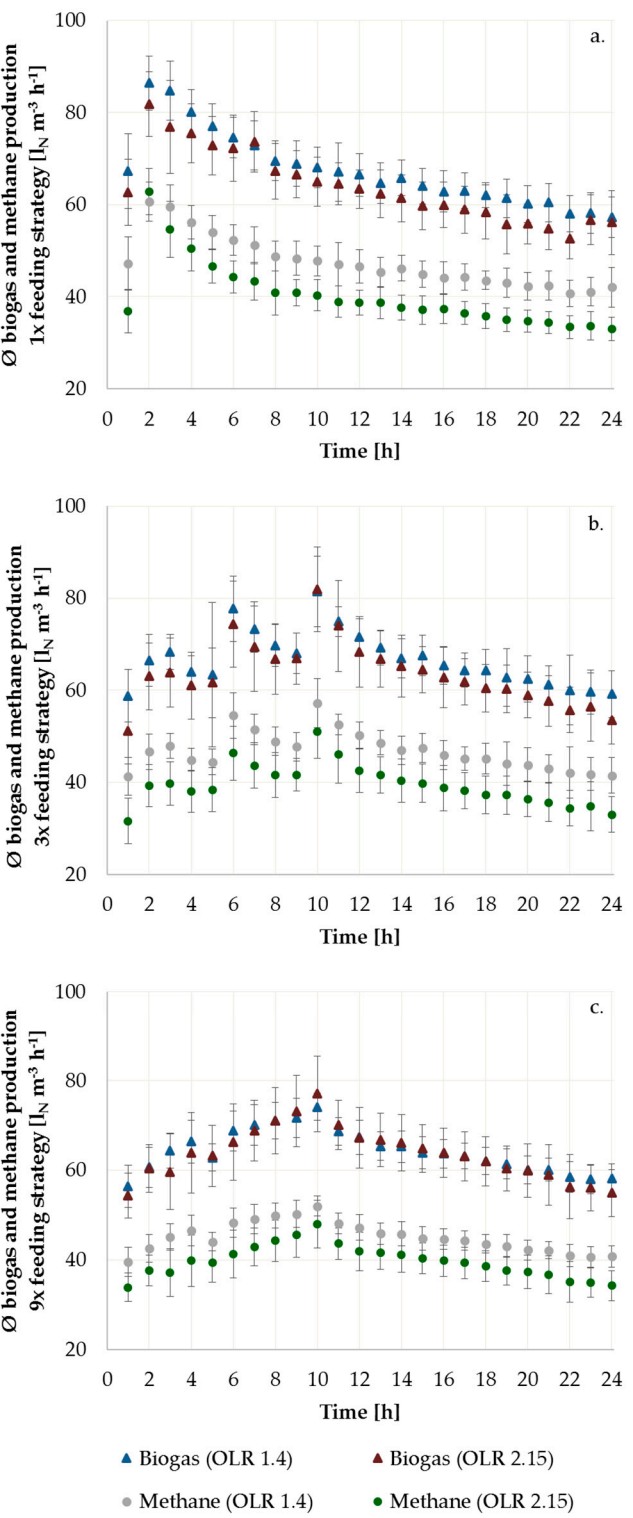

**Figure 3.** Average hourly biogas and methane production $[l_N\ m^{-3}\ h^{-1}]$ of the semi-continuous experiment for the 1x (**a**), 3x (**b**) and 9x (**c**) feeding strategies. The data was collected over 16 working days for feedstock 1 (OLR 1.4 kg vs. $m^{-3}\ d^{-1}$) and 34 working days for feedstock 2 (OLR 2.15 kg vs. $m^{-3}\ d^{-1}$). Feedstock 1: 1x: n = 32; 3x, 9x: n = 48; Feedstock 2: 1x, 9x: n = 68, 3x: n = 102.

Heterogeneity of biogas production after the feeding events was observed between the three feeding strategies, which could result from a different response of microorganisms to the applied feeding quantity. However, regarding the accumulated biogas production, all feeding strategies

showed roughly the same amounts of biogas at the end of the day. This may imply that under the investigated conditions, the feeding events could be shifted to another time (earlier or later) without negatively affecting the overall productivity. The 9x feeding strategy, which represents a conventional biogas performance, showed a more continuous, but slow increase of biogas production than the 1x or 3x feeding strategies. Conversely, the latter two strategies recorded a larger amount of biogas production after the feeding event(s) and a smaller amount during the non-feeding period. These findings might indicate possible suitability for flexible biogas production in practice.

Similar findings regarding more/less frequently-fed strategies were achieved by Mauky et al. and Mulat et al. [7,8]. They reported on differences in biogas production after the feeding events based on the feeding strategy, but the accumulated biogas production remained approximately equal, which is comparable to the present study. Additionally, a sharp increase in biogas production after the feeding events for the less frequently-fed strategies was also observed in these studies, although the kinetic characteristics, as well as the number of feedstocks fed into fermenters, were the determining factors [8]. Eltrop et al. [41] examined the consequences of an intermittent feeding in full scale. The results show peaks of 140–150% of the average daily biogas production, which was achieved by feeding with maize silage three times per day. Peaks of 130–140% of the average daily biogas production resulted from two feeding times (per day) of grain. Similar peaks (140–150%) were achieved by Mauky et al. [8]. These peaks from the similar findings are considerably higher than those from our experiments (1x feeding: 128%, 3x feeding: 122% and 129%, und 9x feeding: 115% and 121% for feedstock 1 and 2, respectively). One explanation could be the slower degradability of feedstocks with a high raw fat content, which was used in this study. The high raw fat content may be very effective (high amounts of biogas and methane production), but the degradability of feedstocks with an easily degradable composition (e.g., high content of sugar, starch, pectin, proteins) is much faster.

Table 2 indicates the average daily biogas and methane production from the semi-continuous test of feedstock 1 and 2 at the three feeding strategies (feeding 1x, 3x and 9x per day). The daily biogas production from the semi-continuous experiment ranged from 1524-1,623 $l_N$ $m^{-3}d^{-1}$, with a methane production of 947-1,138 $l_N$ $m^{-3}$ $d^{-1}$, considering both feedstocks and strategies. Similar results were obtained from the Bruck an der Leitha biogas plant. However, a very high standard deviation in the full-scale performance was measured. This arose from the daily variation in the feedstock composition, which is followed by a diverging OLR. Thus, comparing the volume-specific biogas production rate between full scale and laboratory results seems to be difficult.

**Table 2.** Average daily biogas and methane production [$l_N$ $m^{-3}$ $d^{-1}$] from the semi-continuous experiment Figure 1. 3x and 9x feeding strategies. Feedstock 1: 1x: n = 32; 3x, 9x: n = 48; Feedstock 2: 1x, 9x: n = 68, 3x: n = 102; Bruck an der Leitha biogas plant: n = 365.

| Sample | Biogas Production [$l_N$ $m^{-3}$ $d^{-1}$] | | | Methane Production [$l_N$ $m^{-3}$ $d^{-1}$] | | |
|---|---|---|---|---|---|---|
| | 1x | 3x | 9x | 1x | 3x | 9x |
| Feedstock 1 | 1623 ± 60 | 1590 ± 80 | 1542 ± 31 | 1138± 48 | 1114 ± 57 | 1080 ± 26 |
| Feedstock 2 | 1551 ± 95 | 1524 ± 137 | 1528 ± 118 | 966 ± 61 | 947 ± 95 | 952 ± 75 |
| Biogas plant Bruck an der Leitha (Ø 2015) | 1480 ± 273 | | | 932 ± 175 | | |

The results from the 1x feeding strategy demonstrate the highest average daily biogas and methane production, followed by the results from the 3x feeding strategy. This applies for feedstocks 1 and 2, respectively. The 9x feeding strategy shows the lowest gas production. However, the gas production rates for all feeding strategies were very close to each other. In other words, the average daily gas production indicates roughly the same amounts, which were slightly higher for feedstock 1. The mean values may suggest that the fewer fermenters are fed, the higher the biogas and methane production is. Nevertheless, the results from the t-test for the mean biogas and methane production per day show no

significant differences between the three feeding strategies for both feedstocks ($p < 0.05$). Conversely, the t-test ($p < 0.05$) for the peak values per hour (biogas and methane) demonstrate a high significance between the feeding strategies, which applies to both feedstocks. This implies that production peaks in biogas and methane can be shifted to another time via changing feeding intervals.

The results of our experiment confirm the assumptions of Mauky et al. [7,8] that a flexible feeding regime causes no significant reduction in biogas production. This contradicts the previous assumptions that only continuous feeding can ensure stability [42,43]. Mulat et al. [7] describe two potential factors that may explain a stable biogas and methane production under a flexible feeding regime. First, they note the ability of microorganisms to adapt to changing conditions; second, an extended retention time enables the organic substances to degrade more efficiently. Both factors could also be valid for the semi-continuous experiments in this work.

According to the gas composition, the methane content was around 70 vol.-% for feedstock 1 and 64 vol.-% for feedstock 2 (Table 3).

**Table 3.** Analysis of gas composition from feedstocks 1 and 2 (Ø) (Feedstock 1: n = 24 per strategy for $CH_4$, $CO_2$, $H_2S$; n = 21 per strategy for $H_2$. Feedstock 2: n = 34 per strategy).

| Sample | Feeding Strategy | $CH_4$ | $CO_2$ | $H_2$ | $H_2S$ |
|---|---|---|---|---|---|
| | | [vol.-%] | | [ppm] | |
| Feedstock 1 | 1x | 69.6 ± 0.6 | 30.4 ± 0.6 | 554 ± 127 | 93 ± 20 |
| | 3x | 70.1 ± 0.9 | 29.9 ± 0.9 | 476 ± 67 | 92 ± 10 |
| | 9x | 69.9 ± 0.7 | 30.1 ± 0.7 | 214 ±167 | 45 ± 32 |
| Feedstock 2 | 1x | 64.2 ± 0.9 | 35.8 ± 0.9 | 359 ± 80 | 90 ± 9 |
| | 3x | 64.3 ± 1.2 | 35.7 ± 1.2 | 138 ± 117 | 30. ± 30 |
| | 9x | 64.8 ± 1.0 | 35.2 ± 1.0 | 242 ±122 | 55 ±29 |

Feedstock 1 obtained a 7% higher methane concentration than feedstock 2. This can be explained by the high amount of raw fat contained in feedstock 1 (Table 1) [44,45]. The methane content level within the three feeding strategies from both feedstocks was approximately the same with no significant difference. The $H_2$ concentrations show a large variance between the different feeding strategies; this applies to both feedstocks, whereas the highest concentrations were observed with the 1x feeding strategy. According to Speece [46], an $H_2$ concentration greater than 100 ppm can influence the biogas process negatively, whereas Drosg [42] defines a risk at concentrations of more than 500 ppm. However, given the high methane production rate, no inhibition of the methanogenesis over the acetoclastic path was observed. The $H_2S$ concentration of both feedstocks was lower than the average $H_2S$ concentration of the biogas plant Bruck an der Leitha in the year 2015 (139 ppm), which applies to all feeding strategies.

*3.2. Process Parameter*

3.2.1. Short Chain Fatty Acids

In this experiment, the profile of short chain fatty acids (SCFAs), such as acetic acid, propionic acid, butyric acid, isobutyric acid, valeric acid, isovaleric acid, hexanoic acid and heptanoic acid was analyzed.

According to a guideline from the Bavarian State Research Center for Agriculture (LfL, 2008), the concentration of acetic acid should not exceed 3000 mg $L^{-1}$ FM and 1000 mg $L^{-1}$ FM for propionic acid in a healthy process. An exceedance of these concentration levels might affect anaerobic digestion. As shown in Table 4, the concentrations of both acetic acid and propionic acid were considerably below these values.

**Table 4.** The concentration of acetic acid and propionic acid (mg kg$^{-1}$ FM) in the semi-continuous digestion processes. The results represent the data from feedstock 1 and 2.

| Sample | Measurement Period [days] | Acetic Acid [mg kg$^{-1}$ FM] | | | Propionic Acid [mg kg$^{-1}$ FM] | | |
|---|---|---|---|---|---|---|---|
| | | 1x | 3x | 9x | 1x | 3x | 9x |
| Feedstock 1 | 20 | 926 | 1002 | 1149 | 43.9 | 37.5 | 38.9 |
| | 30 | 553 | 865 | 1373 | <0.03 | 11.1 | 53.6 |
| | 45 | 658 | 1642 | 741 | <0.03 | 64.9 | 31.1 |
| | 58 | 919 | 1118 | 1392 | 15.5 | 20.8 | 43.4 |
| Feedstock 2 | 175 | 1307 | 1406 | 1817 | 11.1 | 50.1 | 43.2 |
| | 180 | 1152 | 1314 | 1651 | 11.1 | 52.9 | 53.1 |
| | 186 | no value | 1038 | 1265 | <0.10 | 23.3 | 31.6 |

Regarding acetic acid, the concentrations fluctuated within the different feeding strategies and accumulated slightly at the end of the measurement period, as compared to their initial values. Although feedstock 1 represented a higher amount of raw fat (around 73% DM), the higher SCFAs were obtained from the fermenters using feedstock 2. Isobutyric acid and isovaleric acid were occasionally detected in the feedstocks, but their concentrations were very low and mostly below the limit of detection (data not shown). Furthermore, SCFAs, such as butyric acid, valeric acid, hexanoic acid and heptanoic acid, could not be detected. According to the results, less frequent feeding did not result in an accumulation of SCFAs. The methanogenic community was capable of converting the intermediates immediately in the given management scenario.

3.2.2. pH Value

The pH value in all fermenters was 7.7 at the beginning of the experiment and increased slightly after starting the feeding process throughout all strategies (Figure 4).

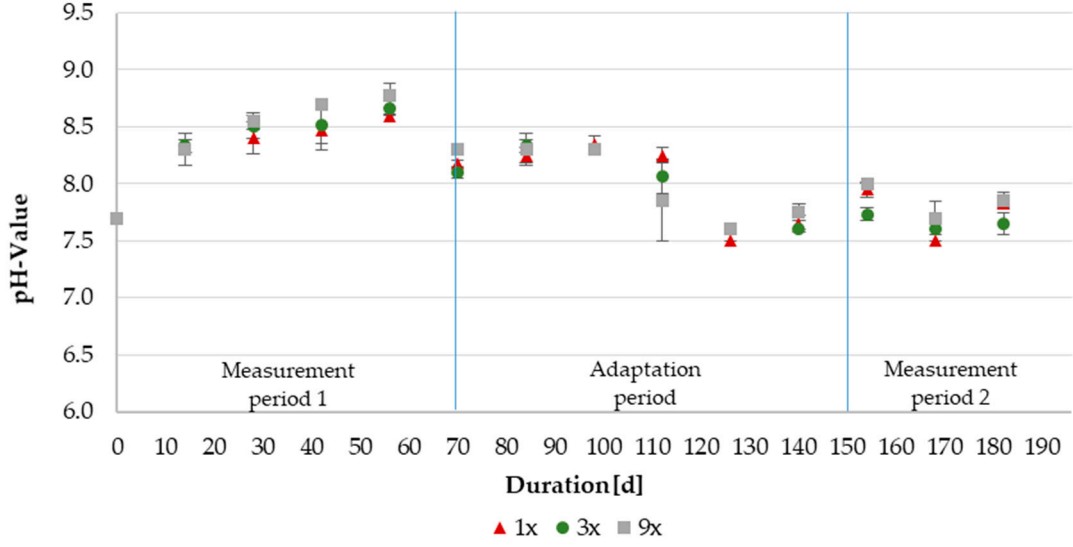

**Figure 4.** The profile of pH values for the three different feeding strategies during the semi-continuous experiment. The total duration of the operation was 186 days. (1x strategy: n = 28, 3x and 9x strategy: n = 42).

The fermenter pH values during measurement period 1 ranged from 8.2 to 8.9. During the adaptation phase, the pH levels fluctuated between 7.5 and 8.4 because of changes to the feedstock and the applied OLRs. During measurement period 2, the fermenter pH varied from 7.5 to 8.0. A pH range between 7.0 and 8.5 is usually essential for anaerobic digestion, while a range from 6.5 to 7.2 is

considered the favorable condition for growing methanogenic microorganisms [47,48]. According to Hansen et al. [21], a pH value greater than eight can favor the conversion of ammonium to ammonia, which supports the inhibition of the anaerobic digestion. However, no negative influence on the process stability was observed.

### 3.2.3. TVA/TIC Ratio

An important indicator of process stability is the TVA/TIC ratio. TVA indicates the volatile organic acids contained in the feedstock, while TIC represents the total organic carbonate [49]. In this experiment, the TVA/TIC ratio was initially at 0.4 and subsequently fluctuated between 0.2 and 0.8 throughout the entire experiment (Figure 5).

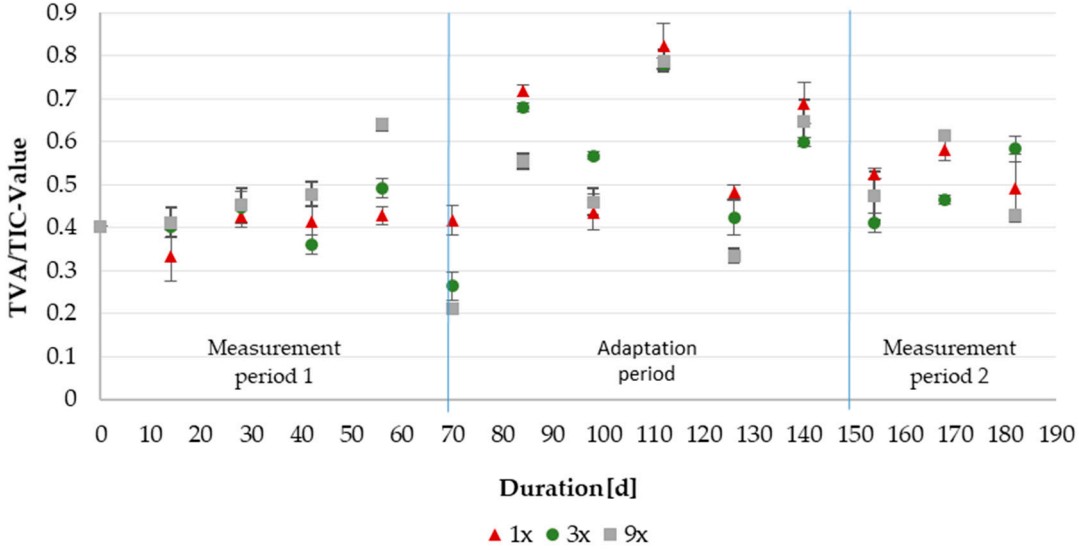

**Figure 5.** Profile of the TVA/TIC ratios for the three different feeding strategies during the semi-continuous experiment. The total duration of the operation was 186 days (1x strategy: n = 28, 3x and 9x strategy: n = 42).

The TVA/TIC ratio of 0.4 indicated that the biogas reactor was not overloaded. A TVA/TIC ratio higher than 0.4 typically indicates unsuitable conditions, such as acidification for the microbial community and can lead to decreased biogas production [32,50,51]. Although the TVA/TIC ratios were frequently at a higher level during the entire experiment, no negative effect on the stability of the anaerobic digestion was observed within all feeding strategies. Thus, a process disturbance—such as acidification—was not recorded. The average TVA/TIC value for the 1x feeding strategy was slightly higher (0.51 ± 0.14) than the values for the 3x (0.49 ± 0.13) and the 9x (0.49 ± 0.14) feeding strategy.

### 3.2.4. Total Ammonium-Nitrogen ($NH_4^+$-N) and Ammonia ($NH_3$-N) Content

An additional buffer capacity for the methanogenic anaerobic digestion was provided by the ammonium/ammonia system in the reactor. Fermenter samples were analyzed to explore the profiles of ammonium and ammonia concentration in this flexible feeding experiment. Feedstock 1 contained 0.49 g $NH_4^+$-N $kg^{-1}$ FM and feedstock 2 0.47 g $NH_4^+$-N $kg^{-1}$ FM. The $NH_4^+$-N (measured) and $NH_3$-N (calculated) contents of the fermenters are described in Table 5. The results demonstrate that the $NH_4^+$-N contents were stable in all fermenters, whereas increased $NH_3$-N concentrations were calculated for the fermenters fed with feedstock 1. These were mostly due to the higher pH value, but also because of the higher $NH_4^+$-N input (up to 0.5 g $L^{-1}$). For the fermenters fed with feedstock 2, no increase of $NH_3$-N was calculated. Between the three feeding strategies, no significant differences in the ammonium and ammonia content were observed.

**Table 5.** Contents of ammonium-nitrogen ($NH_4^+$-N) and ammonia-nitrogen ($NH_3$-N) of the anaerobic digestion from the three different feeding strategies. The samples were collected and analysed during the measurement period from both feedstocks (feedstock 1: 1x: n = 12, 3x,9x: n = 18; feedstock 2: 1x, 9x: n = 12, 3x: n = 18).

| Sample | Measurement Period | Ammonium-Nitrogen ($NH_4^+$-N) | | | Ammonia-Nitrogen ($NH_3$-N) | | |
|---|---|---|---|---|---|---|---|
| | [days] | [mg kg$^{-1}$ FM] | | | [mg L$^{-1}$ FM] | | |
| | | 1x | 3x | 9x | 1x | 3x | 9x |
| Feedstock 1 | 0 | 1200 | 1200 | 1200 | 88 | 88 | 88 |
| | 29 | 1230 | 1250 | 1300 | 170 | 211 | 220 |
| | 48 | 1330 | 1240 | 1230 | 225 | 255 | 305 |
| Feedstock 2 | 158 | 1390 | 1240 | 1390 | 65 | 73 | 42 |
| | 168 | 1340 | 1300 | 1280 | 26 | 32 | 39 |
| | 184 | 1330 | 1340 | 1360 | 40 | 64 | 41 |

The $NH_4^+$-N content described in this study was lower than the concentrations found in the mesophilic stress reactors of previous studies [52,53]. Regarding, the $NH_3$-N concentration, Hansen et al. [21] reported that a concentration of around 1,100 mg L$^{-1}$ caused inhibition of anaerobic digestion from swine manure. Lebuhn et al. [9] described ammonia toxicity, which was occasionally observed above 500 mg L$^{-1}$ (with continuous feeding, particularly at a higher OLR). However, in this study, the $NH_3$-N concentrations are much lower than the values stated above. Therefore, no negative effects on process stability caused by excessive ammonia concentration were found.

### 3.2.5. Molecular Biology Analyses

The methanogenic archaea constitute the most sensitive microbial guilds in biomethanation processes. Molecular process diagnosis and early warning parameters, such as the T/G$_{mcr A1,2}$ ratio and the metabolic quotient (MQ) (see Section 2.6.) specifically tackling this microbial group by qPCR with *mcr*A$_{1,2}$ DNA and cDNA had been developed for mesophilic digestion of maize silage [9,12]. These ecophysiological parameters were recently evaluated also for validity at different process temperatures and with different feedstock components [54].

Figure 6 shows the concentration of *mcr*A$_{1,2}$ DNA and cDNA along with the T/G$_{mcr A1,2}$ ratios (cDNA/DNA) and the MQ values for the three feeding strategies with the two feedstocks. For the samples fed with feedstock 1, *mcr*A$_{1,2}$ cDNA and DNA concentrations were not significantly different among the three feeding strategies and likewise not dissimilar to those fed 1x and 3x per day with feedstock 2. Only the sludges in the fermenters fed 9x per day with feedstock 2 showed slightly higher cDNA and DNA concentrations. Nevertheless, the *mcr*A$_{1,2}$ cDNA and DNA concentrations in the fermenter sludges fed with both substrates were all in the normal range for efficient biogas-producing biocenoses at the given OLRs [9].

The T/G$_{mcr A1,2}$ (or *mcr*A$_{1,2}$ cDNA/DNA) ratio depicts the mRNA copies produced per methanogenic cell, indicating the specific transcriptional activity of methanogens in a given sample. The T/G$_{mcr A1,2}$ ratios all ranged between 0.1 and 1. They were at best slightly higher for feedstock 2 than for feedstock 1 (Figure 6). Values between 0.01 and 1 are considered normal for sludges producing biogas efficiently [9], indicating no major impact of the treatments and the differences between the process conditions on the activity of the methanogens.

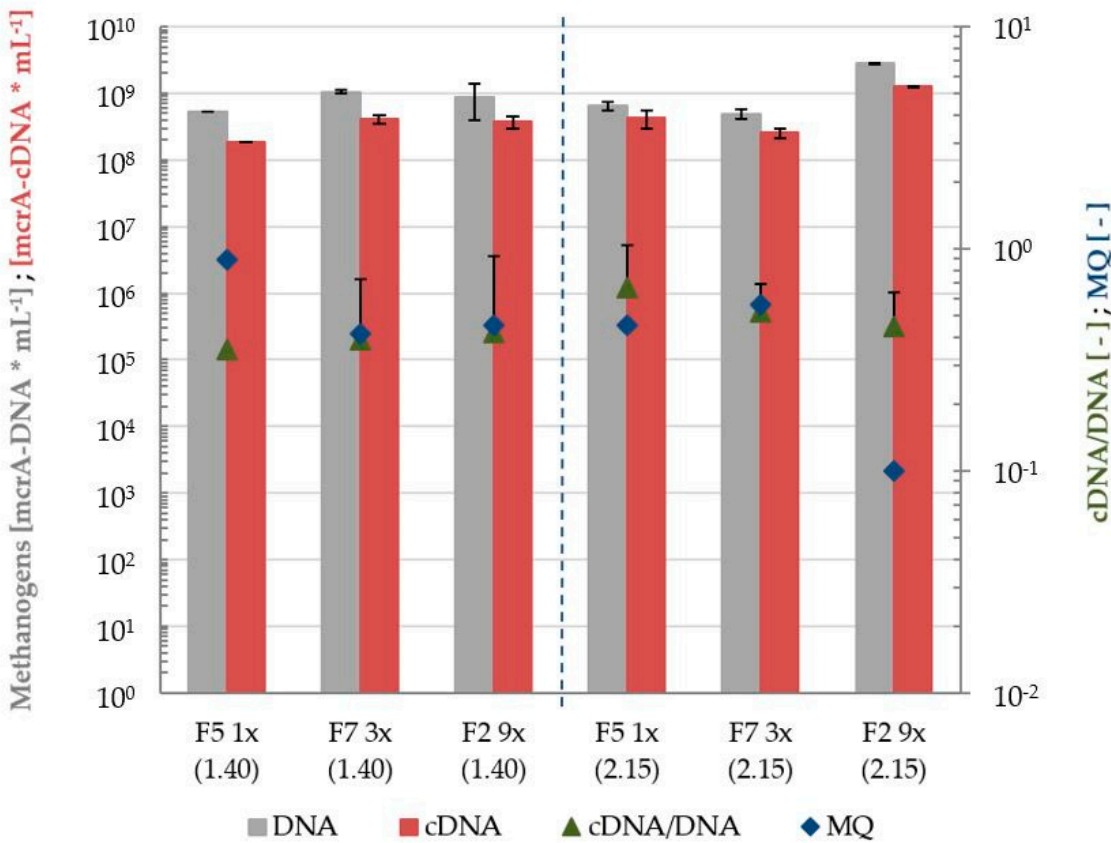

**Figure 6.** $mcrA_{1,2}$ DNA and cDNA concentrations, $T/GmcrA_{1,2}$ ratios and metabolic quotient (MQ) values in the differently fed fermenter sludges.

The MQ is defined as the ratio of the specific methanogenic activity (SA) of the methanogenic cells at a given time ($SA_{act}$) to the predicted specific activity ($SA_{pred}$). $SA_{pred}$ is calculated by an empirical standard function using the actual methane productivity at points in time without symptoms of process disturbance over a broad range of OLRs and the concentration of methanogens present at that time [9,12]. MQ values around 1 indicate standard specific methanogenic activity. MQ values between 0.1 and 3–4 are in the normal range [12]. A high MQ indicates metabolic strain or even stress for the methanogens, whereas low MQ values can be a sign either of under-challenge, or of distress—possibly leading to process breakdown [55].

All MQ values were in the lower normal range, but there was a tendency towards lower MQ values with higher feeding frequency for both feedstocks (Figure 6), indicating slightly lower strain and a slightly higher loading capacity reserve in these variants. Accordingly, slightly higher TVA/TIC ratios were obtained for the less frequently-fed fermenters, indicating a marginally higher acid accumulation caused by feeding more substrate at a single event and, consequently, a slightly higher metabolic strain of the methanogenic associations, as depicted by the slightly higher MQ values.

Profound changes to the microbial community composition can be expected upon changing the feeding regime, and especially upon changing the substrate. Radical changes to the feedstock composition are reported to cause profound structural community modification—possibly leading to process disturbance, whereas by switching between more similar, conventionally-used substrates appears to evoke minor alterations within the microbial community [56–58]. Therefore, only minor structural changes are expected in the current study. High microbial diversity in the sludges—warranting high functional redundancy—obviously had prevented process disturbance, as suggested by the process monitoring data and the results from the molecular ecophysiological analyses. The seemingly adequate adaptation between operation periods with the different substrates may have further smoothed the substrate switch impacts on the systems' performance.

### 3.3. Effects on Biogas Storage Capacities

The obtained biogas and methane production curves for the two investigated feedstocks and three flexible feeding strategies were implemented in the process simulation model of the biogas plant. The three power generation scenarios described in Section 2.7 were simulated, and the reduction potential of the needed biogas storage capacity compared to continuous feeding was determined. Figures 7 and 8 show the biogas storage levels during the simulated power generation scenarios for feedstock 1 (OLR 1.4 kg vs. m$^{-3}$ d$^{-1}$) and feedstock 2 (OLR 2.15 kg vs. m$^{-3}$ d$^{-1}$), respectively.

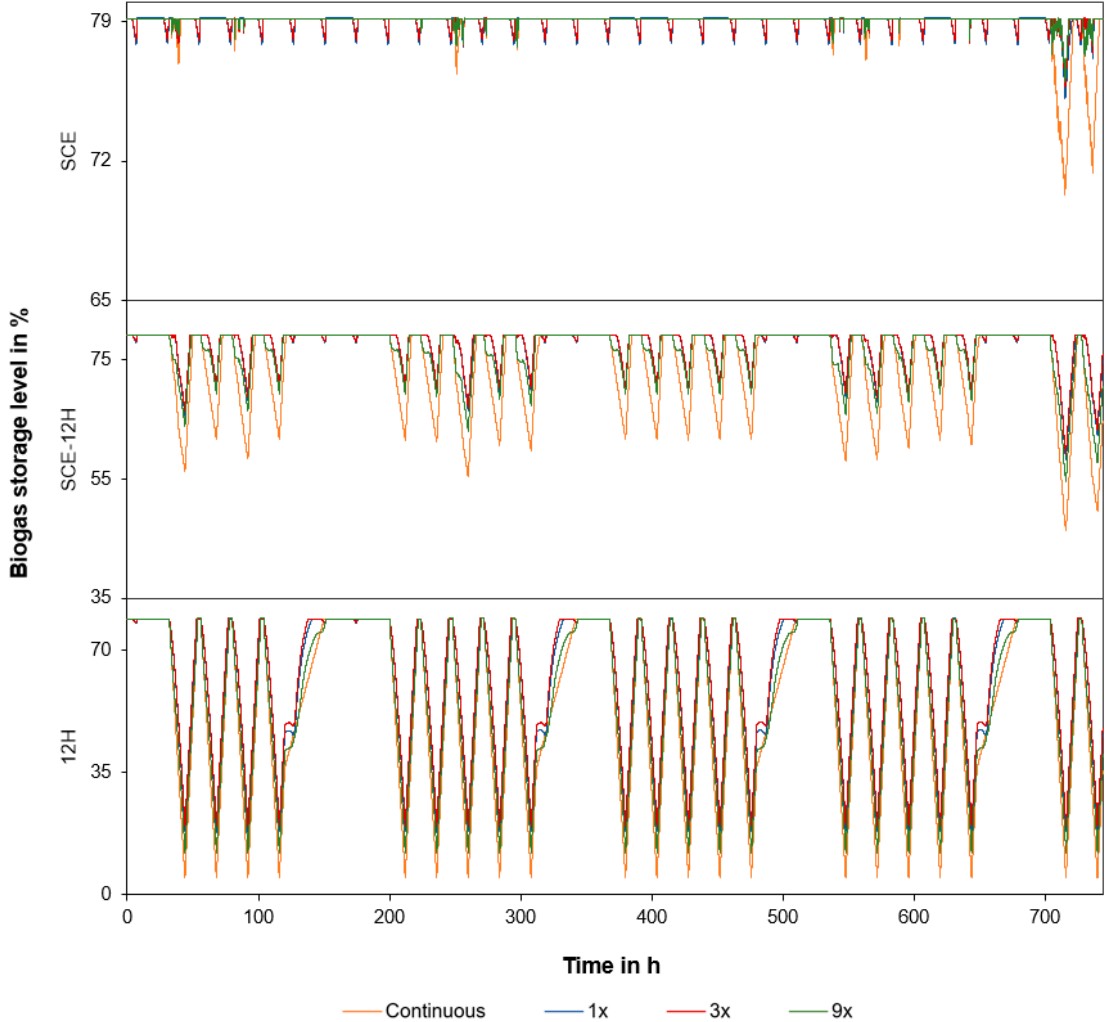

**Figure 7.** Comparison of the three investigated feeding strategies to continuous feeding of feedstock in terms of simulated biogas storage level for OLRs of 1.4 kg vs. m$^{-3}$ d$^{-1}$.

Continuous feeding was compared to the three investigated feeding strategies (1x, 3x and 9x per day) for two feedstock mixtures with different OLRs (1.4 and 2.15 kg vs. m$^{-3}$ d$^{-1}$). The results for scenarios SCE and SCE-12H show that the needed biogas storage capacity could be reduced by demand-oriented feeding for all of the three investigated feeding strategies in comparison to continuous feeding.

The results from scenario 12H and continuous feeding reveal for both OLRs that the installed biogas storage capacity at the biogas plant was not sufficient to provide enough biogas for power generation in peak times while continuously producing biomethane (the minimum biogas storage level of 5% was reached, which led to negative and unreasonable mass flow results). However, the biogas storage capacity proved sufficient for all of the investigated feeding regimes at an OLR of 1.4 kg vs. m$^{-3}$ d$^{-1}$ and when feedstock was fed once per day with an OLR of 2.15 kg vs. m$^{-3}$ d$^{-1}$. These

results indicate that certain power generation strategies that cannot be executed when the feedstock is fed continuously, due to limited biogas storage capacity may be executable when the feeding regime is changed. Table 6 summarizes the reduction potential of the needed biogas storage capacity for the simulated scenarios.

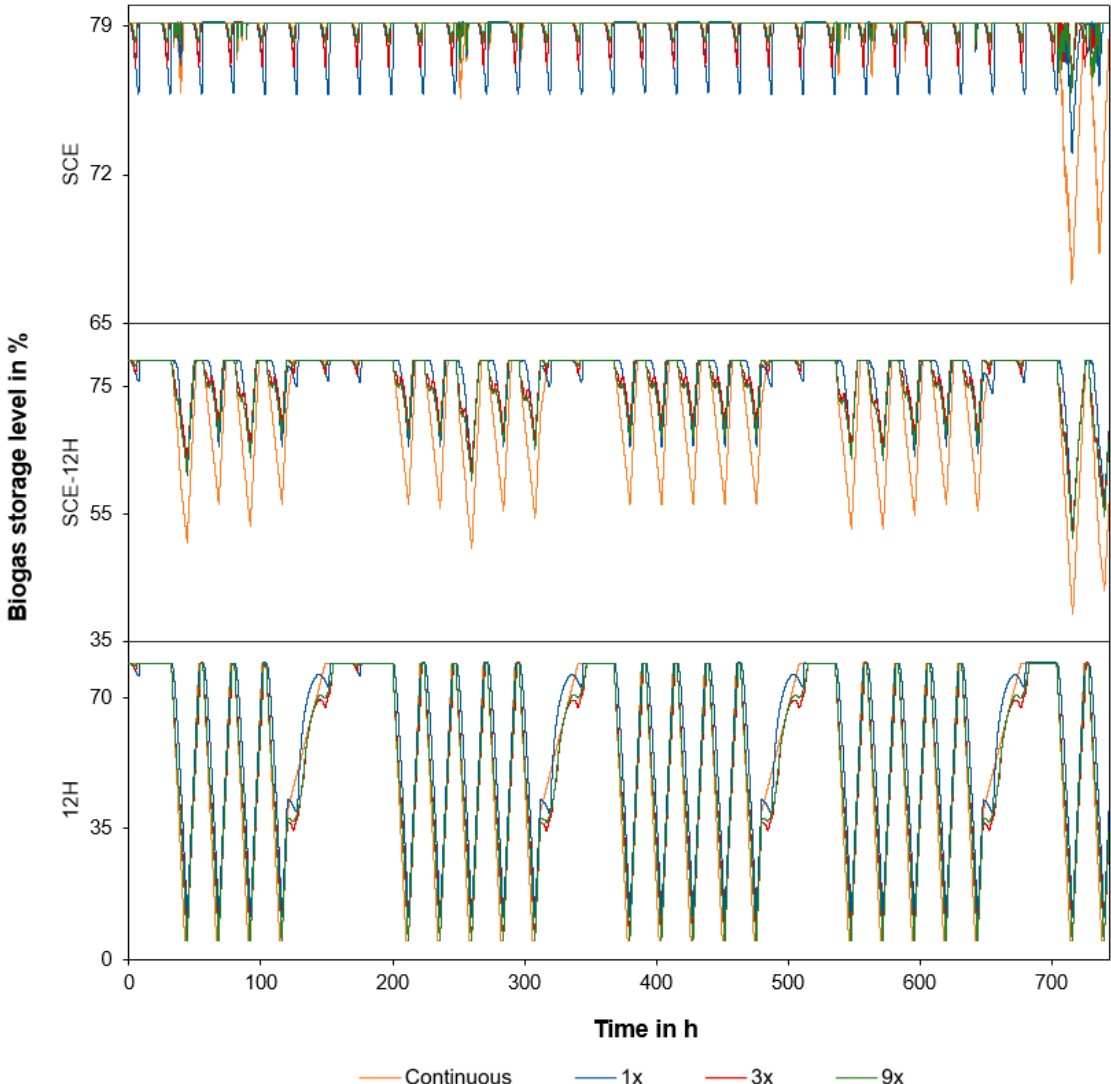

**Figure 8.** Comparison of the three investigated feeding strategies to continuous feeding of feedstock in terms of simulated biogas storage level for OLRs of 2.15 kg vs. m$^{-3}$ d$^{-1}$.

Biogas storage investment costs significantly contribute to the biogas plant's economic viability. A biogas storage volume reduction of several hundred cubic meters leads to savings of tens of thousands of Euros if compared to common cost functions in literature, e.g., in Lauer et al. [59].

In this study, the highest biogas storage reduction potential with 73.7% was observed for scenario SCE, when feedstock was fed 3x per day with an OLR of 2.15 kg vs. m$^{-3}$ d$^{-1}$. This difference in biogas storage level between continuous and flexible feeding occurred during the end of the simulated period, when control energy reserves were activated exceptionally frequently (more than fifty times in the simulated last week). There was hardly any difference between continuous and flexible feeding strategies during the rest of the simulated period, except for small fluctuations in the biogas storage level that occurred during flexible feeding strategies.

For scenario SCE-12H biogas storage savings range between 25.2–38.9%. Similar values were reported in other studies that assessed demand-oriented feeding strategies. Mauky et al. [5]

demonstrated that the needed biogas storage capacity could be reduced by up to 45% within the flexible power generation context. Barchmann et al. [17] reported biogas storage saving potentials of up to 65.1% for a weekly optimized power generation schedule and results from Grim et al. [13] showed that biogas storage investment costs could be reduced by 24–53% by flexible feeding strategies.

**Table 6.** The reduction potential of needed biogas storage capacity for investigated power generation scenarios and feeding regimes.

| Scenario | Feeding Regime | Feedstock 1 | | Feedstock 2 | |
|---|---|---|---|---|---|
| | | Needed Biogas Storage Volume | Reduction Potential | Needed Biogas Storage Volume | Reduction Potential |
| | | [m³] | [%] | [m³] | [%] |
| SCE | Continuous | 425 | - | 591 | - |
| | 1x | 194 | 54.3 | 299 | 49.5 |
| | 3x | 166 | 60.9 | 156 | 73.7 |
| | 9x | 155 | 63.7 | 161 | 72.8 |
| SCE-12H | Continuous | 1579 | - | 1916 | - |
| | 1x | 1004 | 36.4 | 1270 | 33.7 |
| | 3x | 964 | 38.9 | 1295 | 32.4 |
| | 9x | 1181 | 25.2 | 1353 | 29.4 |
| 12H | Continuous | 3571 | - | 3571 | - |
| | 1x | 2940 | 17.7 | 3266 | 8.5 |
| | 3x | 2865 | 19.8 | 3571 | - |
| | 9x | 3234 | 9.5 | 3571 | - |

## 4. Conclusions

Three different feeding strategies (feeding 1x, 3x and 9x per day) were applied to assess their effects on the process performance of a semi-continuous biogas system under mesophilic conditions. In this experiment, two different types of feedstock—one with high fat content (feedstock 1) and the other with high carbohydrate content (feedstock 2)—were used. The performance of biogas and methane production was analyzed hourly and daily. It can be concluded that the feeding intervals of 1x and 3x feeding per day can simulate a flexible biogas system. Furthermore, a stable biogas and methane production were reported throughout all strategies, whereas the 1x feeding strategy achieved the highest amounts of biogas, followed by the 3x and 9x feeding strategy, but showed no significant difference (according to the t-test with $p < 0.05$). Conversely, a high significance has been shown for the values with the highest-achievable biogas and methane production rate per hour (both feedstocks, all feeding strategies), which implies that biogas and methane production peaks can be shifted to another time via changing feeding intervals. This applies to all feeding strategies, as well as both feedstocks.

Another finding was that the less frequent feeding strategies had no negative impacts on the overall process performance. The microbial analysis suggested that changing the feedstock did not severely affect the specific activity of methanogenic archaea.

The results from the semi-continuous fermentation tests (biogas production curves and composition) were implemented in a process simulation model of biogas storage and utilization at the Bruck an der Leitha biogas plant. Three different scenarios focusing on power generation during peak times were simulated, and the needed biogas storage capacity was calculated. The results showed that the needed storage capacity could be reduced by up to 73.7% compared to continuous feeding if flexible feeding strategies are applied. Furthermore, flexible feeding strategies enabled some power generation strategies that could not be executed while feeding input material continuously, due to insufficient biogas storage capacity.



The flexibility and demand-oriented power generation with biogas plants are both crucial in serving the needs of the future energy system. The findings of this study showed that it is possible to supply biogas flexibly at the investigated biogas plant with a focus on waste treatment without any negative impacts on process stability. This could lead to considerable economic savings in the context of demand-oriented power generation, due to reductions of needed biogas storage capacity. The results of this study could help in invalidating reservations of biogas plants operators against applying flexible feeding strategies. However, it is strongly recommended to assess the limits of flexible feeding by fermentation tests in order to secure process stability.

**Author Contributions:** Conceptualization, E.S., S.F., A.B.; methodology, A.B., E.S., L.M., S.F.; software, A.M.; validation, E.S.; formal analysis, S.F., A.B., J.L. (Javier Lizasoain), J.L. (Jonas Leber), K.K., B.M., L.M.; investigation, E.S., K.K., J.L. (Javier Lizasoain), J.L. (Jonas Leber), B.M., S.F.; resources, A.M., A.F., A.B., A.G., L.M.; data curation, J.L. (Javier Lizasoain), J.L. (Jonas Leber), K.K., S.F., B.W.; statistics; S.F.; writing—original draft preparation, E.S., S.F., M.L., K.K.; writing—review and editing, A.M., A.B., M.L.; visualization, E.S., S.F., K.K., B.M., M.L., A.B.; supervision, A.M., A.F., A.B., A.G.; project administration, A.M.; funding acquisition, A.B., A.M., S.F.

**Funding:** This research was funded by the Austrian Research Promotion Agency (FFG), grant number D16621150400.

**Acknowledgments:** The publication is supported by BOKU Vienna Open Access Publishing Fund. Major parts of this study were conducted and funded over the course of the Bio(FLEX)Net project (project number D16621150400) that was supported by the Austrian Research Promotion Agency (FFG). The authors want to thank the operators of the Bruck an der Leitha biogas plant Bernadette Mauthner and Gerhard Danzinger for providing information about their plant and feedstock samples.

**Conflicts of Interest:** The authors declare no conflict of interest.

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
