# Peer review of "Utilization of Food and Agricultural Residues for a Flexible Biogas Production: Process Stability and Effects on Needed Biogas Storage Capacities"

_energies, doi:10.3390/en12142678_

Round 1

Reviewer 1 Report

I appreciate the complexity and high quality of presented research paper and have only minor comments and suggestions such are:

·       Line 145 – I would suggest to add % methane share in the produced biogas while presenting the total biogas production.

·       Line 158 – The term “titration method” is rather used than titrimetric method.

·       Line 213 – It is not clear, what the authors mean the term “analysed biomass”. Please, provide specifications.

·       Chapter 2.4 – Based on the text, the selection of types of feedstock (feedstock 1 – high in fat and feedstock 2 – high in carbohydrate) and their serial feeding (feedstock 1 first, then feedstock 2) is not fully clear. Lines 227-228 provide only partial explanation on OLR “The OLR applied in this experiment were simulated from the operating system of the biogas plant at Bruck an der Leitha.“ Please, provide more detail information about the above-mentioned. If the feedstock selection and feeding is not in accordance with that in the biogas plant at Bruck an der Leitha, the serial feeding might influence the results and parallel experiments might be more suitable for overall comparison.

·       Table 3 – Even though, values less than 100ppm, a large variance of H2S concentrations are shown, but unfortunately not discussed in the text. It would be interesting for the readers to provide an explanation over these results.

·       Line 569 – correct double dots at the end of the sentence;

·       Lines 610, 614, 617 – correct the units of OLR.

Author Response

Dear Editor, Dear reviewer, 

Thank you very much for your comments. We have shortened the introduction of the manuscript. We have changed all the points and hope that we have fulfilled all your expectations. Enclosed you will find the revised version. All changes (excluding deletions) are highlighted in yellow.

Ad  Line 145 – add % methane share added

Ad Line 158  term used “titration method” 

Ad  Line 213 – Information added

Ad Chapter 2.4 – Information added in Chapter 2.2.1

Ad Table 3 – Sentence added in the document

Ad  Line 569 – done

Ad  Lines 610, 614, 617 – done

Alexander Bauer 

Reviewer 2 Report

In this study, the authors present the results obtained with three different feeding strategies of two different feedstocks on the process performance of a semi-continuous biogas system.

Although there are several studies in literature which present similar studies, all contributions are needed for further developments in this area. The work appears scientifically sound and results are presented in a clear manner.

I only suggest that authors should short the manuscript in particularly Introduction.

Author Response

Dear Editor, Dear reviewer, 

Thank you very much for your comments. We have shortened the introduction of the manuscript. Enclosed you will find the revised version. 

Alexander Bauer 
